# Control Variates for Slate Off-Policy Evaluation

**Nikos Vlassis**
Netflix

**Ashok Chandrashekar**[*]
WarnerMedia

**Fernando Amat Gil**
Netflix

**Nathan Kallus**
Cornell University and Netflix

## Abstract

We study the problem of off-policy evaluation from batched contextual bandit data with multidimensional actions, often termed slates. The problem is common to recommender systems and user-interface optimization, and it is particularly challenging because of the combinatorially-sized action space. Swaminathan et al. (2017) have proposed the pseudoinverse (PI) estimator under the assumption that the conditional mean rewards are additive in actions. Using control variates, we consider a large class of unbiased estimators that includes as specific cases the PI estimator and (asymptotically) its self-normalized variant. By optimizing over this class, we obtain new estimators with risk improvement guarantees over both the PI and the self-normalized PI estimators. Experiments with real-world recommender data as well as synthetic data validate these improvements in practice.

## 1 Introduction

Online services (news, music and video streaming, social media, e-commerce, app stores, etc.) serve content that often takes the form of a combinatorial, high-dimensional *slate*, where each slot on the slate can take multiple values. For example, a personalized news service can have separate slots for local news, sports, politics, etc., with multiple news items from which to select in each slot (Li et al., 2010); or a video streaming service may organize the contents of the landing homepage of a user as a multi-slot slate, where each slot can contain shows from a given category (Gomez-Uribe and Hunt, 2015). Typically, the offered content is personalized via machine learning and A/B testing. However, the number of eligible items per slot may be in the hundreds or thousands, which makes the testing and optimization of personalized policies a challenging task.

An alternative to A/B testing and online learning is *off-policy evaluation* (OPE). In OPE we use historical data collected by a past policy in order to evaluate a new candidate policy. This is primarily an estimation problem, often grounded in causal assumptions about the data-generating process and its connections to a counterfactual one (Horvitz and Thompson, 1952; Robins and Rotnitzky, 1995; Dudík et al., 2011; Bottou et al., 2013; Athey and Wager, 2017; Bibaut et al., 2019; Kallus and Uehara, 2019). However, even with plentiful off-policy data, handling a combinatorial action space can be challenging: Unbiased importance sampling (IS) (Horvitz and Thompson, 1952) suffers variance on the scale of the cardinality of the action space under a uniform logging policy and deterministic target policy. To address this, Swaminathan et al. (2017) have proposed the pseudoinverse (PI) estimator, which significantly attenuates the variance and remains unbiased for decomposable reward structures. In this paper, we discuss how to further reduce this variance using a *control variates* approach (Glynn and Szechtman, 2002). We demonstrate that the self-normalized PI (wPI) estimator, which Swaminathan et al. (2017) found to be superior to PI, is asymptotically equivalent to adding a control variate. But, the choice of control variate is not optimal. We instead show how to choose optimal

---

[*]The author was with Netflix when this work was concluded.

35th Conference on Neural Information Processing Systems (NeurIPS 2021).

control variates, and even how to do so without incurring any bias. We provide strong empirical evidence based on the MSLR-WEB30K data from the Microsoft Learning to Rank Challenge (Qin and Liu, 2013) that confirms the improvement of our approach over the PI and wPI estimators.

## 1.1 Related work

Much of the existing OPE work involves the use of *doubly-robust* estimators and variants, which add control variates to the standard IS estimator (Robins and Rotnitzky, 1995; Dudík et al., 2011; Thomas and Brunskill, 2016; Farajtabar et al., 2018; Bibaut et al., 2019; Kallus and Uehara, 2019; Su et al., 2020). Directly optimizing control variates for OPE has been suggested earlier (Farajtabar et al., 2018; Vlassis et al., 2019) but not for problems involving high-dimensional slates.

Combinatorial bandits and variants such as semi-bandits have been studied by Cesa-Bianchi and Lugosi (2012) and Kveton et al. (2015) and are also covered in the book of Lattimore and Szepesvári (2020). Earlier approaches to OPE for contextual combinatorial bandits have been restricted to small slate problems (Li et al., 2011) or relied on accurate parametric models of slate rewards (Chapelle and Zhang, 2009; Guo et al., 2009; Filippi et al., 2010). Li et al. (2015) and Wang et al. (2016) have suggested the use of partial matches between the logging and target actions in order to mitigate the variance explosion. Swaminathan et al. (2017) have proposed the PI estimator, a version of which we study in this paper. Under factored policies, the PI estimator is similar to an estimator proposed by Li et al. (2018) for OPE of ranking policies with user click models. Swaminathan and Joachims (2015) and Joachims et al. (2018) have developed estimators for OPE under the framework of counterfactual risk minimization and discuss approaches for offline policy optimization. Agarwal et al. (2018) have developed an OPE method for rankers, when only the relative value of two given rankers is of interest. Chen et al. (2019) have developed an approach for offline policy gradients, when slates are unordered lists of items (of random size). McInerney et al. (2020) and Lopez et al. (2021) have studied the problem of slate OPE with semi-bandit feedback (one observed reward per slot), under different assumptions on the structure of the rewards and the contextual policies. In constrast to the latter two works, and similar to Swaminathan et al. (2017) and Su et al. (2020), we study the slate OPE problem when only a single, slate-level reward is observed. Su et al. (2020) achieve variance reduction by clipping propensities, whereas we achieve variance reduction by optimizing control variates.

Applications of slate OPE in real-world problems include Sar Shalom et al. (2016) who apply OPE to a large scale real world recommender system that handles purchases from tens of millions of users, Hill et al. (2017) who present a method to optimize a message that is composed of separated widgets (slots) such as title text, offer details, image, etc, and McInerney et al. (2020) who apply OPE to a slate problem involving sequences of items such as music playlists.

## 2 Setup and Notation

We consider contextual slate bandits where each slate has $K$ slots. We will use $[K]$ to denote the set $\{1, 2, \ldots, K\}$. The available actions in slot $k$ are $[d_k] = \{1, \ldots, d_k\}$. Thus, the set of available slates has cardinality $\prod_{k=1}^{K} d_k$. The data consists of $n$ independent and identically distributed (iid) triplets $(X_i, A_i, R_i)$, $i = 1, \ldots, n$, representing the observed context, slate taken, and resulting reward, respectively. Here, $A_i = (A_{i1}, \ldots, A_{iK})$ where $A_{ik} \in [d_k]$. We use $(X, A, R)$ to refer to a generic draw of one of the above triplets, where $X \sim \mathbb{P}(X), A \sim \mu(\cdot \mid X), R \sim \mathbb{P}(\cdot \mid A, X)$. The distribution $\mu(a \mid x) = \mathbb{P}(A = a \mid X = x)$ is called the *logging policy* as the data can be seen as the logs generated by executing this policy. Probabilities $\mathbb{P}$, expectations $\mathbb{E}$, and (co)variances without subscripts indicating otherwise are understood to be with respect to this distribution. For any function $f(x, a, r)$ we will write $\mathbb{E}_n[f(X, A, R)] = \frac{1}{n} \sum_{i=1}^{n} f(X_i, A_i, R_i)$. We will also follow the convention that *random, data-driven* objects have the form $\hat{\cdot}_n$ (such as $\hat{w}_n$ or $\hat{\theta}_n$).

In OPE, we are given a fixed policy $\pi(a \mid x)$ and are interested in estimating its average reward $\mathbb{E}_\pi[R]$ using data collected under a logging policy $\mu(a \mid x)$. Here $\mathbb{E}_\pi$ refers to expectation under the distribution $X \sim \mathbb{P}(X), A \sim \pi(\cdot \mid X), R \sim \mathbb{P}(\cdot \mid A, X)$. The quantity $\mathbb{E}_\pi[R]$ can be interpreted as a counterfactual given that $\mathbb{P}(\cdot \mid A = a, X)$ is equal to the distribution of the *potential* reward of slate $a$ given context $X$, for every $a$; or, in other words, that $\mathbb{P}(\cdot \mid A, X)$ represents a structural model. This assumption, known as unconfoundedness or ignorability, is only needed for giving a causal interpretation to $\mathbb{E}_\pi[R]$ but not for estimation.

The classical IS estimator for the problem of estimating $\mathbb{E}_\pi[R]$ is $\mathbb{E}_n\left[\frac{\pi(A|X)}{\mu(A|X)}R\right]$, which is unbiased but can suffer unacceptably large variance when $A$ is high-dimensional and $\mu$ provides coverage of all slates, as is necessary for the identification of arbitrary policies. Generally, we cannot do much better than IS in the worst case (Wang et al., 2017), unless we impose additional assumptions about the structure of actions or rewards.

In this paper, we will focus on the case in which the logging policy is *factored*, that is, $\mu$ satisfies $\mu(a \mid x) = \prod_{k=1}^{K} \mu_k(a_k \mid x)$. This generally holds if $\mu$ is given by a per-slot $\epsilon$-greedy policy or is the uniform distribution, both are common cases in practice.

We define the following slot-level density ratios and their sum

$$Y_k = \frac{\pi(A_k \mid X)}{\mu_k(A_k \mid X)}, \qquad G = 1 + \sum_{k=1}^{K}(Y_k - 1). \tag{1}$$

Then, we have the following reformulation result for $\mathbb{E}_\pi[R]$. (A succinct proof of this result, as well as all our novel results, can be found in Section 9 of this paper and in the Supplementary Text.)

**Lemma 1** (Reformulation under additive rewards (Swaminathan et al., 2017))**.** *Under a factored $\mu$, $E_\pi[R] = \mathbb{E}[GR]$ whenever $\mathbb{E}[R \mid A, X] = \sum_{k=1}^{K} \phi_k(A_k, X)$ for some (latent) functions $\phi_k(a_k, x)$.*

Motivated by this result, in this paper we focus on estimating the target parameter $\theta = \mathbb{E}[GR]$. Crucially, lemma 1 implies that under additive rewards, the variance of an estimator of $\theta$ need only suffer the size of the slot-level density ratios (namely, $\sum_{k=1}^{K} d_k^2$ for uniform exploration and deterministic evaluation) rather than the possibly astronomical size of the slate density ratio appearing in the IS estimator (namely, $\prod_{k=1}^{K} d_k^2$ under the same settings). Moreover, lemma 1 shows that $\mu$ need not even cover every slate combination – only marginally every action per slot. In the rest of the paper we will not assume any special structure for $\mathbb{E}[R \mid A, X]$; our estimation results on $\theta$ will hold irrespective of whether $\theta$ coincides with $E_\pi[R]$. In the experiments we will show results for both additive and non-additive rewards.

In the factored policy setting, the PI and wPI estimators of Swaminathan et al. (2017) are, respectively,

$$\hat{\theta}_n^{\mathrm{PI}} = \mathbb{E}_n[GR], \qquad \hat{\theta}_n^{\mathrm{wPI}} = \frac{\mathbb{E}_n[GR]}{\mathbb{E}_n[G]}.$$

The PI estimator is unbiased for estimating $\theta$ by construction. While wPI may be biased, it can further reduce PI's variance.

## 3   A Class of Estimators

We now construct a class of unbiased estimators that includes PI and (almost) wPI. We will then propose to choose *optimal* estimators in this class and show how they can be approximated.

For any fixed tuple of weights $w = (w_1, \ldots, w_K) \in \mathbb{R}^K$, we define the estimator

$$\hat{\theta}_n^{(w)} = \mathbb{E}_n[\Gamma_w], \qquad \Gamma_w = GR - \sum_{k=1}^{K} w_k(Y_k - 1).$$

Here, each $Y_k - 1$ acts as a *control variate* (Glynn and Szechtman, 2002). We slightly overload notation and for the case $w_1 = \cdots = w_K = \beta \in \mathbb{R}$ we will write

$$\hat{\theta}_n^{(\beta)} = \mathbb{E}_n[\Gamma_\beta], \qquad \Gamma_\beta = GR - \beta(G - 1),$$

which corresponds to using a single control variate $G - 1$. Note that $\hat{\theta}_n^{\mathrm{PI}}$ is included in the estimator class: When $\beta = 0$, we have $\Gamma_0 = GR$ and $\hat{\theta}_n^{\mathrm{PI}} = \hat{\theta}_n^{(0)}$. We further define $V_w = \mathrm{Var}(\Gamma_w)$. It is straightforward to show that, for any *fixed* $w$, the estimator $\hat{\theta}_n^{(w)}$ is unbiased (for any $n$) and asymptotically normal.

**Lemma 2** (Unbiasedness and asymptotic normality of $\hat{\theta}_n^{(w)}$)**.** *For any fixed $w$, we have*

$$\mathbb{E}[\hat{\theta}_n^{(w)}] = \mathbb{E}[\Gamma_w] = \theta,$$
$$\sqrt{n}(\hat{\theta}_n^{(w)} - \theta) \to_d \mathcal{N}(0, V_w).$$

While PI is in this class and has variance $V_0$, wPI is not actually a member of this class, but it is asymptotically equivalent to a member. To see this, we first need to understand the asymptotics of $\hat{\theta}_n^{(w)}$ when we plug in a *random, data-driven* $w$ denoted $\hat{w}_n$. It turns out (by Slutsky's theorem) that, if $\hat{w}_n$ converges to some *fixed* vector $w$, the asymptotic behavior of $\hat{\theta}_n^{(\hat{w}_n)}$ is the same as plugging in the fixed $w$.

**Lemma 3** (Asymptotics of plug-in $w$). *Suppose $\hat{w}_n \to_p w$, for some fixed $w$. Then*

$$\sqrt{n}(\hat{\theta}_n^{(\hat{w}_n)} - \theta) \to_d \mathcal{N}(0, V_w).$$

We can now establish the following corollary showing that wPI is asymptotically equivalent to $\hat{\theta}_n^{(\theta)}$, that is, a member of our class of estimators using $\beta = \theta$ (the unknown target estimand).

**Lemma 4** (Asymptotics of wPI). *We have*

$$\sqrt{n}(\hat{\theta}_n^{wPI} - \theta) \to_d \mathcal{N}(0, V_\theta).$$

# 4 Optimal Control Variates

In the previous section we showed that both PI and wPI are, or are asymptotically equivalent to, members of the class of estimators $\hat{\theta}_n^{(w)}$. But the class is more general than these two, so it behooves us to try to find an optimal member.

## 4.1 Optimal Single Control Variate

We first focus on the case $w_1 = \cdots = w_K = \beta$, that is, using a single control variate $G - 1$. Both PI and wPI fall in this category with $\beta = 0$ and $\beta = \theta$, respectively, as we showed above. Let

$$V^\dagger = \inf_{\beta \in \mathbb{R}} V_\beta \tag{2}$$

represent the minimal variance in the class of single-control-variate estimators. By construction, $V^\dagger \le V_0 = \text{Var}(GR)$ and $V^\dagger \le V_\theta$, the right-hand sides being the (asymptotic) variance of PI and wPI, respectively. Our next result derives this optimum.

**Lemma 5** (Optimal single control variate). *We have*

$$V^\dagger = V_{\beta^*} = V_0 - \frac{(\mathbb{E}[G^2 R] - \theta)^2}{\sum_k \text{Var}(Y_k)},$$

*where*

$$\beta^* = \frac{\mathbb{E}[G^2 R] - \theta}{\sum_k \text{Var}(Y_k)}.$$

Notice that generally $\beta^* \ne \theta$. That is, while wPI is asymptotically equivalent to a control-variate estimate, it is not generally optimal. In the degenerate case where $R \perp\!\!\!\perp G$ (*e.g.*, constant reward) then we do have $\beta^* = \theta$.

Our results, nonetheless, immediately suggest how to obtain this optimum asymptotically, using a feasible estimator.

**Lemma 6** (Achieving optimal single control variate). *Compute a data-driven $\beta$ as*

$$\hat{\beta}_n^* = \frac{\mathbb{E}_n[GR(G - 1)]}{\sum_k \mathbb{E}_n[(Y_k - 1)^2]}.$$

*Then,*

$$\sqrt{n}(\hat{\theta}_n^{(\hat{\beta}_n^*)} - \theta) \to_d \mathcal{N}(0, V^\dagger).$$

That is, asymptotically, $\hat{\theta}_n^{(\hat{\beta}_n^*)}$ is at least as good as either PI and wPI. The improvement is positive, in general.

**Lemma 7.** *The improvement of $\hat{\theta}_n^{(\hat{\beta}_n^*)}$ over PI is*

$$V_0 - V^\dagger = \frac{\mathbb{E}[GR(G-1)]^2}{\mathbb{E}[(G-1)^2]} \geq 0,$$

*and the improvement over wPI is*

$$V_\theta - V^\dagger = \frac{\mathbb{E}[GR(G-1)]^2}{\mathbb{E}[(G-1)^2]} - 2\mathbb{E}[GR]\mathbb{E}[G^2R] + \mathbb{E}[GR]^2(2 + \mathbb{E}[(G-1)^2]) \geq 0.$$

## 4.2 Optimal Multiple Control Variates

We now turn our attention to optimally choosing weights on all $K$ control variates. Let

$$V^* = \inf_{w \in \mathbb{R}^K} V_w. \tag{3}$$

By construction, $V^* \leq V^\dagger \leq \min(V_0, V_\theta)$, that is, it is smaller than or equal to the asymptotic variances of PI, wPI, and the optimal single-control-variate estimator. Empirically, we find that the first inequality can have a small gap while the second a large one; that is, optimizing control variates provides substantive improvement over PI and wPI, but much of the improvement is often captured by optimizing a single control variate in certain highly symmetric settings. See sections 6 and 7.

**Lemma 8** (Optimal multiple control variates). *We have*

$$V^* = V_{w^*} = \sum_{k=1}^K \frac{\mathbb{E}[GR(Y_k - 1)]^2}{\mathrm{Var}(Y_k)},$$

*where*

$$w_k^* = \frac{\mathbb{E}[GR(Y_k - 1)]}{\mathrm{Var}(Y_k)}.$$

Again, if $R \perp\!\!\!\perp Y_k$, we have $w_k = \theta$ for all $k$. Generally, $w_k$ vary over $k$ and are not equal to $\theta$.

As in the single control-variate case above, we can obtain the above optimum asymptotically using a feasible estimator.

**Lemma 9** (Achieving optimal multiple control variates). *Compute data-driven weights $w_k$ as*

$$\hat{w}_{n,k}^* = \frac{\mathbb{E}_n[GR(Y_k - 1)]}{\mathbb{E}_n[(Y_k - 1)^2]}.$$

*Then,*

$$\sqrt{n}(\hat{\theta}_n^{(\hat{w}_n^*)} - \theta) \rightarrow_d \mathcal{N}(0, V^*).$$

**Lemma 10.** *The improvement of $\hat{\theta}_n^{(\hat{w}_n^*)}$ over PI is*

$$V_0 - V^* = \sum_k \frac{\mathbb{E}[GR(Y_k - 1)]^2}{\mathbb{E}[(Y_k - 1)^2]} \geq 0.$$

*The improvement over the single control variate $\hat{\theta}_n^{(\hat{\beta}_n^*)}$ is*

$$V^\dagger - V^* = \sum_k \frac{\mathbb{E}[GR(Y_k - 1)]^2}{\mathbb{E}[(Y_k - 1)^2]} - \frac{\mathbb{E}[GR(G-1)]^2}{\mathbb{E}[(G-1)^2]} \geq 0.$$

This improvement is of course no smaller than the improvement over wPI, which is no better than the optimal single control variate (asymptotically). The improvement again collapses to zero in the special case of constant rewards.

## 5  Achieving Optimality Without Suffering Bias

Although the above estimators have asymptotically optimal variance among control variates, they may incur finite-sample bias due to the estimation of control variate weights. We can avoid this using a three-way cross-fitting. The estimator we construct in this section will be *both* finite-sample unbiased *and* have optimal variance asymptotically.

The estimator we propose is as follows:

1. Split the data randomly into three even folds, $\mathcal{D}_0, \mathcal{D}_1, \mathcal{D}_2$.
2. For $j = 0, 1, 2$, compute $\hat{w}_{n,k}^{(j)} = \frac{\mathbb{E}_{\mathcal{D}_j}[GR(Y_k - 1)]}{\mathbb{E}_{\mathcal{D}_j}[(Y_k - 1)^2]}$, where $\mathbb{E}_{\mathcal{D}_j}$ refers to the sample average only over $\mathcal{D}_j$.
3. Set

$$\hat{\theta}_n^* = \mathbb{E}_n[GR] - \sum_k \sum_{j=0,1,2} \frac{|\mathcal{D}_j|}{n} \hat{w}_{n,k}^{(j+1 \bmod 3)} \mathbb{E}_{\mathcal{D}_j}[Y_k - 1].$$

**Lemma 11** (Unbiased estimator with optimal asymptotic variance)**.** *We have*

$$\mathbb{E}[\hat{\theta}_n^*] = \theta, \qquad \sqrt{n}(\hat{\theta}_n^* - \theta) \to_d \mathcal{N}(0, V^*).$$

While using just two folds would suffice to eliminate bias, ensuring that each data point $i$ is independent from the data used to fit its weight, this would not suffice for ensuring we get the optimal variance. For this, it is crucial that we use three folds so as to eliminate covariance as well. Having three folds ensures that, given any two data points $i$ and $i'$, either $i$ is disjoint from the data used to fit the weights of $i'$ or vice versa. That is, for any two folds $j$ and $j'$, we always have either $j + 1 \neq j' \pmod 3$ or $j' \neq j + 1 \pmod 3$.

In summary, lemma 11 establishes that we can completely avoid bias even in finite samples. We could in fact adapt our cross-fitting procedure to obtain alternative estimators that are both finite-sample unbiased and match the asymptotic variance of the single-control-variate version or even wPI; for that we would need only change the weights computed in step 2. We note, however, that the variance characterization in all of our results is asymptotic and we do not currently have corresponding finite-sample results.

## 6  Experiments on real data

We have benchmarked the proposed estimators on the publicly available dataset MSLR-WEB30K from the Microsoft Learning to Rank Challenge (Qin and Liu, 2013). This is a labeled dataset that contains about 31k user queries, each providing up to 1251 labeled documents (the labels are relevance scores from 0 to 4). The queries form the contexts $x$ of our OPE problem. In order to be able to provide a head-to-head comparison with the results reported by Swaminathan et al. (2017), we have closely followed the experimental protocol of the latter (with minor differences, explained next) in order to generate contextual slate bandit instances from the data. (Additional details about the MSLR-WEB30K data and the experimental protocol that we followed can be found in the Supplementary Text.)

Each query-document pair in the dataset is annotated with 'title' and 'body' features. As a preprocessing step, we first use all data to train a regression tree that predicts document scores from the 'title' features of query-document pairs. We then use this regressor as a standard greedy ranker to extract, for each query, the top-$M$ predicted documents (with $M \in \{10, 50, 100\}$). Finally, we discard all queries that have less than $M$ judged documents. (This latter step was not present in the experimental pipeline of Swaminathan et al., 2017. We include it here because it facilitates sampling with replacement, which is required in our experiments. Even for $M = 100$, the resulting dataset contains a sizable number (18k+) of unique queries, and running the PI and wPI estimators against these data attained essentially the same performance as with the unfiltered dataset.)

The above procedure defines the sets $A(x)$ of 'allowed' documents per context $x$. In each problem instance, we train a new predictor (using decision trees or lasso) constrained on the sets $A(x)$, and use it as a greedy ranker to extract, for each query, the top-$K$ predicted documents (with $K \in \{5, 10, 30\}$). This mapping defines a deterministic target policy $\pi(x)$ for our OPE problem: each $K$-size ranking

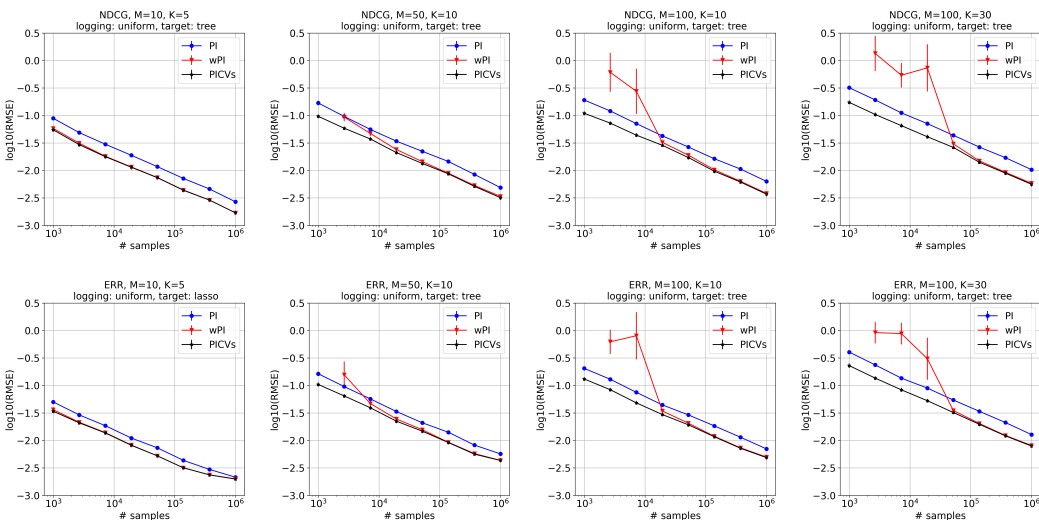

Figure 1: Benchmarking the proposed PICVs estimator against PI and wPI on the MSLR-WEB30K data, using a tree-based regression model for the target policy. Here we vary $M$, the number of available actions (documents) per slot, $K$, the number of slots (size of ranked lists), and the choice of metric that defines slate-level reward. Missing values for wPI (for sample size 1k) are due to excessive variance caused by the presence of outliers. See text for details.

defines a $K$-slot slate, with each slot having cardinality $M$. Note that each document can appear only once in the slates mapped by $\pi(x)$. The logging policy $\mu(x)$ is uniform, and it samples $K$ documents *with replacement* from the set $A(x)$, for each context $x$. (In Swaminathan et al., 2017, documents were sampled without replacement by the logging policy.) To create each logged dataset, we sampled the $x$ uniformly from the set of all non-filtered queries, up to the desired sample size (which, as in Swaminathan et al., 2017, could end up reusing some queries multiple times).

We report results for PI, wPI, and the optimal single control variate estimator $\hat{\theta}_n^{(\hat{\beta}_n^*)}$ from lemma 6, denoted PICVs. The multiple control variate estimator from lemma 9 obtained near-identical MSE as the reported PICVs, so we omit it from the plots. (The similar performance of the single vs multiple control variate estimators in this problem is owed to the symmetry between slots; for example, each having the same number of actions. In the next section we show an example where the two estimators exhibit different behavior.) We evaluate each estimator using (log) root mean square error (RMSE) as a function of sample size. We estimate MSE and standard errors of $\log(\text{RMSE})$ (the latter computed via the delta method on $\frac{\log(\text{MSE})}{2\log(10)}$) using 300 independent runs for each setting. As in Swaminathan et al. (2017), we repeat the protocol for a number of different experimental conditions, varying the values of $M, K$, the regression model for the target policy (tree-based or lasso), and the choice of metric. We tested two metrics, NDCG (additive) and ERR (non-additive); see Swaminathan et al. (2017) for definitions. The version of NDCG that we used is tailored to rankings with replacements: It only differs to standard NDCG in that the denominator is the DCG of a slate that is formed by the globally most relevant document (for the given $x$) replicated over all $K$ slots of the slate (capturing the fact that documents are sampled with replacement in $\mu$). Note that this version of NDCG is additive over slots, which is a requirement for PI (and, asymptotically, for our PICVs estimator) to be unbiased when the estimand is the value of the target policy. The code for these experiments is publicly available at https://github.com/fernandoamat/slateOPE.

In Fig. 1 we show the results for the NDCG and ERR metrics when using a tree-based predictor to define the target policy. (See Supplementary Text for results when using lasso.) We observe that the proposed estimator PICVs dominates both PI and wPI, in both metrics, with notable improvements for small sample sizes relative to $M$.

For relatively small sample sizes, wPI would often return abnormally large estimates (values up to four orders of magnitude larger than the median). This can happen when the denominator of wPI gets, by chance, close to zero. This in an artifact of wPI being defined as the ratio of estimates and

can lead to very high variance—even higher than PI's. Our proposed PICVs is not prone to such outliers and the ensuing variance, and is robust uniformly over all the sample sizes tested.

While wPI seems to converge to PICVs for large sample sizes, this need not be the case in general. Lemma 4 shows that wPI has asymptotic variance $V_\theta$, that is, wPI is equivalent to using the weight $\beta = \theta$ on the single control variate. Since, by definition, $V_\theta \geq V^\dagger = V_{\beta^*}$, it is not true in general that wPI should converge to PICVs, at least in the sense of having similar asymptotic distributions, unless certain special conditions hold so that these variances are equal. The estimators do appear similar for large sample sizes in Fig. 1 due to the nature of the particular dataset and the reported setup (choice of metric, target policy, and value of $K$); see Supplementary Text for different problem settings, where we can observe a uniform gap between PICV and (w)PI.

Our theoretical results support the observation that the MSE of the various estimators in Fig. 1 wil asymptotically appear as parallel curves that never catch up. Note that the plots in Fig. 1 are on a log-log scale (log(MSE) vs log(# samples)). Our theory predicts that, for large enough sample sizes $n$, the MSE of the different estimators will be $V/n$ for different values of $V$. Since $\log(V/n) = \log(V) - \log(n)$, we expect all methods to eventually appear linear in the log-log scale with the very same slope (hence parallel), but at different heights given by the corresponding $V$ (hence never catch up). The fact that PICVs is always at the bottom is consistent with lemma 6, which shows that PICVs attains the smallest asymptotic variance among all single-control-variate estimators.

# 7 Experiments on synthetic data: The gap between PICV single and multi

The results on the MSLR-WEB30K data do not demonstrate any substantial differentiation between the single and multi variants of the proposed PICV estimators. This is mainly due to symmetry between slots, which need not always be the case. In this section we show results from synthetic simulations on a non-contextual slate bandit problem, using a reward model that is skewed toward the first slot of the slate and toward the target policy (which picks action 0 from each slot). This setting reveals a differential behavior of the single vs the multi PICV variants. Here we assume Bernoulli slate-level rewards, and the slate-level Bernoulli rates $p(a)$ are given by the weighted sum

$$p(a) = (0.5)^{a_1-1}\phi_1(a_1) + 0.01 \sum_{k=2}^{K} \phi_k(a_k), \tag{4}$$

where the slot-level actions are $a_k \in \{1, \ldots, d_k\}$, for $k = 1, \ldots, K$. Under this model, slot-1 actions that are closer to action 1 are conferring more impact to the slate-level reward, and hence we expect the latter to be more correlated with the target policy. This allows us to examine if there is any differential behavior between the single (PICVs) and the multi (PICVm) variants.

In each simulation experiment, we generate $T = 20$ random reward tensors $p(a) \in [0,1]^D$ using eq. (4), where, for each $a_k$, the value of $\phi_k(a_k)$ is drawn from a Gaussian distribution $\mathcal{N}(0.2/K, 0.01)$. The parameters of each simulated instance are the number $N$ of logged slates, the number of slots $K$, and the slot action cardinality tuple $D = [d_1, ..., d_K]$ where $d_i$ is the number of available actions of the $i$-th slot. For each sampled tensor, we generate 300 datasets by drawing slates $a \sim \mu(\cdot)$ using a uniform logging policy $\mu(a)$ and drawing Bernoulli rewards from the corresponding $p(a)$. We use each dataset to evaluate a deterministic policy $\pi(a) = \mathbb{I}(a = [0, ..., 0])$ (w.l.o.g.) using each of the candidate estimators. Since we know the ground truth, we can compute the MSE of each estimator. We report (log) average root mean square error (RMSE), with standard errors (almost negligible), over the $T$ tensors. The code for these experiments is publicly available at https://github.com/fernandoamat/slateOPE.

The results are shown in Fig. 2 and demonstrate a differentiation between PICVs and PICVm, with the multi variant obtaining lower RMSE by virtue of leveraging more control variates. In the Supplementary Text we show the analogous plot for larger sample sizes.

# 8 Conclusions and Discussion

We studied the problem of off-policy evaluation for slate bandits. We considered a class of unbiased estimators that included PI and (asymptotically) wPI, constructed using a control variate approach. This strongly suggested trying to obtain the minimal variance in this class. We showed how to do so

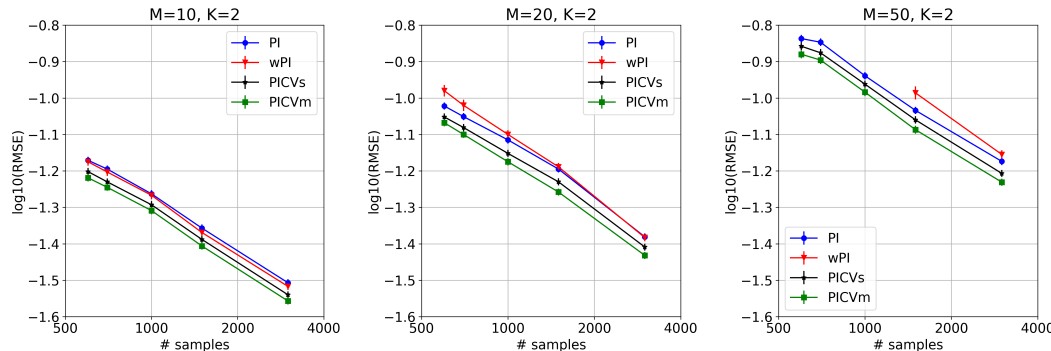

Figure 2: Comparing the estimators in a small scale simulation setting, demonstrating a differentiated behavior between the single and multi variants of PICV. See text for details. Missing values for wPI are due to excessive variance caused by the presence of outliers.

asymptotically, and even how to do so without incurring any additional bias in finite samples. Our new estimators led to gains in both simulations and real-data experiments.

**Future Work.** An interesting avenue for future research is how to meld this approach with doubly-robust (DR) style centering. Su et al. (2020) consider the DR-PI estimator given by adding to the PI estimator the (unweighted) control variate $\mathbb{E}[Gf(X, A) - \sum_{a \in \prod_{k=1}^{K} [d_k]} \pi(a \mid X) f(X, a)]$. In particular, they consider $f(x, a)$ being an estimate of $\mathbb{E}[R \mid A = a, X = x]$. But unlike the standard non-additive OPE case, it is not clear that it would be efficient to consider only this control variate (Kallus and Uehara, 2019). In particular, restricting the statistical model to have additive rewards make standard semiparametric efficiency analyses difficult, and the efficient influence function is unknown and likely has no analytic form. Again, we can consider a range of *centered* control variates, $\sum_{k=1}^{K} \left( \mathbb{E}[Y_k f_k(X, A_k)] - \mathbb{E}[\sum_{a_k \in [d_k]} \pi(a_k \mid X) f_k(X, a_k)] \right)$. Letting $f_k(x, a_k) = w_k$ recovers our estimator class $\hat{\theta}_n^{(w)}$. However, there may be benefit to using $(x, a)$-dependent control variates. Efficiency considerations suggest letting $f_k$ estimate $\mathbb{E}[R \mid A_k = a_k, X = x]$, but the (sub)optimality of such in the additive-rewards model requires further investigation. A promising direction forward is to consider a parametric family, $f_k \in \mathcal{F}_k$, and optimize the choice of parameter to minimize an empirical variance estimate, in the spirit of more robust doubly robust estimation (Farajtabar et al., 2018). As long as $\mathcal{F}_k$ has nice complexity, we can obtain the best-in-class variance asymptotically. Characterizing this precisely and investigating the value of this empirically remains future work.

**Societal Impacts.** In general, having accurate methods for performing OPE can allow decision makers to evaluate potentially unsafe decision making policies, and decide whether they may lead to improved societal outcomes, without having to risk the potentially negative consequences of trialling such policies. Nonetheless, as discussed above, OPE with combinatorial actions is nearly hopeless without making additional assumptions. These, however, can fail to hold and lead to biased estimates, which needs to be taken into account when considering potential harms of a proposed policy. Another potential risk, which applies to OPE in general, is that the data used may not accurately reflect the diversity of the population that the policy will actually be deployed on. In this case, resulting policy value estimates may be much more accurate for individuals who are well represented in the data, compared with individuals who are not. This may bias downstream decision making towards policies that best serve those who are well represented in the data, possibly at the cost of those who are not well represented. Such biases must also be taken into account in any OPE analysis.

# 9 Proofs of the main results

Here we provide succinct proofs for all results, except for lemma 11 whose proof is quite involved and long and is therefore deferred to the Supplementary Text.

*Proof of lemma 1.* Using the definition of $G$ from eq. (1), we have

$$\mathbb{E}[GR \mid A, X] = \sum_{k=1}^{K} Y_k \, \phi_k(A_k, X) + \sum_{k=1}^{K} \left( 1 - K + \sum_{j \neq k} Y_j \right) \phi_k(A_k, X).$$

Taking expectation over $A$, the last term cancels (when $\mu$ is factored) since $\mathbb{E}[Y_k \mid X] = \sum_{a_k \in [d_k]} \pi(a_k \mid X) = 1$, and the first term is $\mathbb{E}_\pi[R \mid X]$. Further taking expectation over $X$ gives the result. $\square$

*Proof of lemma 2.* Since $\mathbb{E}[Y_k \mid X] = 1$, we have $\mathbb{E}[\Gamma_w \mid X] = \mathbb{E}[GR \mid X]$. Iterated expectations gives the first statement. The second statement is immediate from the central limit theorem (CLT). $\square$

*Proof of lemma 3.* Set $\tilde{\theta}_n = \mathbb{E}_n[\Gamma_w]$. From CLT, $\sqrt{n}(\tilde{\theta}_n - \theta) \to_d \mathcal{N}(0, V_w)$. Next, note that

$$\sqrt{n}(\tilde{\theta}_n - \hat{\theta}_n) = \sum_{k=1}^{K} (\hat{w}_{n,k} - w_{n,k}) \sqrt{n} \mathbb{E}_n[Y_k - 1].$$

Since $\sqrt{n} \mathbb{E}_n[Y_k - 1] \to_d \mathcal{N}(0, \mathrm{Var}(Y_k))$ by CLT and $\hat{w}_{n,k} - w_{n,k} \to_p 0$ by assumption, we have by Slutsky's theorem that each of the $K$ terms converges in distribution to the constant 0, and therefore also converges in probability to the constant 0. Hence, $\sqrt{n}(\tilde{\theta}_n - \hat{\theta}_n) \to_p 0$. Applying Slutsky's theorem again establishes the claim of the lemma. $\square$

*Proof of lemma 4.* Since $\mathbb{E}_n[GR] \to_p \theta$ and $\mathbb{E}_n[G] \to_p 1$, the continuous mapping theorem gives $\hat{\theta}_n^{\mathrm{wPI}} \to_p \theta$. Next, note that we can rewrite

$$\frac{\mathbb{E}_n[GR]}{\mathbb{E}_n[G]} = \mathbb{E}_n[GR] + \mathbb{E}_n[GR] \frac{1 - \mathbb{E}_n[G]}{\mathbb{E}_n[G]} = \mathbb{E}_n[GR] - \hat{\theta}_n^{\mathrm{wPI}} \mathbb{E}_n[G-1] = \mathbb{E}_n[GR] - \sum_k \hat{\theta}_n^{\mathrm{wPI}} \mathbb{E}_n[Y_k - 1]$$

and invoking lemma 3 completes the proof. $\square$

*Proof of lemma 5.* We have

$$V_\beta = \mathrm{Var}(GR) - 2\beta \mathbb{E}[GR(G-1)] + \beta^2 \mathbb{E}[(G-1)^2],$$

which is convex in $\beta$. Differentiating, setting to zero, and solving for $\beta$, we get $\beta^* = \frac{\mathbb{E}[GR(G-1)]}{\mathbb{E}[(G-1)^2]}$. For the numerator, we have $\mathbb{E}[GR(G-1)] = \mathbb{E}[G^2 R] - \mathbb{E}[GR] = \mathbb{E}[G^2 R] - \theta$. For the denominator, we have $\mathbb{E}[(G-1)^2] = \mathrm{Var}(G) = \sum_k \mathrm{Var}(Y_k)$. Combining yields the claim of the lemma. $\square$

*Proof of lemma 6.* This is direct from the weak law of large numbers and lemmas 3 and 5. $\square$

*Proof of lemma 7.* This follows by algebra and noting that $V^\dagger \geq \max(V_0, V_\theta)$ since both $\beta = 0$ and $\beta = \theta$ are feasible in eq. (2). $\square$

*Proof of lemma 8.* Let $C = (Y_1 - 1, \ldots, Y_K - 1)$. We have

$$V_w = \mathrm{Var}(GR) - 2w^\intercal \mathbb{E}[GRC] + w^\intercal \mathbb{E}[CC^\intercal]w,$$

which is convex in $w$. Differentiating, setting to zero, and solving for $w$ gives $w^* = \mathbb{E}[CC^\intercal]^{-1} \mathbb{E}[GRC]$. We have $\mathrm{Cov}(Y_k, Y_{k'}) = 0$ whenever $k \neq k'$. Hence we can simplify and obtain the result. $\square$

*Proof of lemma 9.* This is direct from the weak law of large numbers and lemmas 3 and 8. $\square$

*Proof of lemma 10.* This follows by algebra and noting that $V^* \geq \max(V_0, V^\dagger)$ since both $w_1 = \cdots = w_K = 0$ and $w_1 = \cdots = w_K = \beta^*$ are feasible in eq. (3). $\square$

## Acknowledgments

We want to thank Adith Swaminathan for helping with the MSLR-WEB30K data and code, and for many useful discussions.

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
