# Control Variates for Slate Off-Policy Evaluation: Supplementary Text

**Nikos Vlassis**
Netflix

**Ashok Chandrashekar**[*]
WarnerMedia

**Fernando Amat Gil**
Netflix

**Nathan Kallus**
Cornell University and Netflix

## Abstract

In this Appendix we provide additional details about the MSLR-WEB30K data and the experimental protocol that we followed, we prove Lemma 11 of the main paper, and we show additional results on the MSLR-WEB30K and the simulated data.

## A  Details about the MSLR-WEB30K data and the experimental protocol

The MSLR-WEB30K dataset is available for download from https://www.microsoft.com/en-us/research/project/mslr/, is distributed under Standard MSR License Agreement, and does not contain any personally identifiable information.

To train the regression trees models we used sklearn.tree.DecisionTreeRegressor (with parameters criterion = "mse", splitter = "random", min_samples_split = 4, min_samples_leaf = 4), optimized using sklearn.model_selection.GridSearchCV (with main parameters max_depth = 3 and cv = 3). To train the lasso models we used sklearn.linear_model.Lasso (with parameters fit_intercept = False, max_iter = 500, tol = 1e-4, normalize = False, precompute = False, copy_X = False, warm_start = False, positive = False, random_state = None, selection = 'random'), optimized using sklearn.model_selection.GridSearchCV (with main parameters max_depth = 3 and cv = 3). See the provided code for more details.

For all our experiments we employed a total of about 1k quad-core machines on the cloud, each running for about two hours on average.

## B  Proof of Lemma 11

**Lemma 11** (Unbiased estimator with optimal asymptotic variance). *We have*

$$\mathbb{E}[\hat{\theta}_n^*] = \theta,$$
$$\sqrt{n}(\hat{\theta}_n^* - \theta) \to_d \mathcal{N}(0, V^*).$$

*Proof.* For each $j = 0, 1, 2$, we have

$$\mathbb{E}[\hat{w}_{n,k}^{(j+1 \mod 3)}\mathbb{E}_{\mathcal{D}_j}[Y_k - 1]] = \mathbb{E}[\hat{w}_{n,k}^{(j+1 \mod 3)}]\mathbb{E}[\mathbb{E}_{\mathcal{D}_j}[Y_k - 1]]] = 0$$

because the folds are independent. Therefore,

$$\mathbb{E}[\hat{\theta}_n^*] = \mathbb{E}[\mathbb{E}_n[GR]] = \theta,$$

---

[*]The author was with Netflix when this work was concluded.

35th Conference on Neural Information Processing Systems (NeurIPS 2021).

establishing the first statement.

Let $j(i)$ be such that $i \in \mathcal{D}_j$. And, for brevity, define $m(j) = j + 1 \mod 3$. Let $Z = RG$ and $C = (Y_1 - 1, \ldots, Y_K - 1)$. Define

$$U_i = Z_i + (\hat{w}_n^{(m(j(i)))})^{\intercal} C_i.$$

Then $\mathbb{E}[U_i] = \theta$, and for $i \neq i'$:

$$\begin{aligned}
\mathrm{Cov}(U_i, U_{i'}) = {} & \mathbb{E}[Z_i Z_{i'}] - \theta^2 \\
& - \mathbb{E}[Z_i (\hat{w}_n^{(m(j(i')))})^{\intercal} C_{i'}] \\
& - \mathbb{E}[Z_{i'} (\hat{w}_n^{(m(j(i)))})^{\intercal} C_i] \\
& + \mathbb{E}[(\hat{w}_n^{(m(j(i)))})^{\intercal} C_i (\hat{w}_n^{(m(j(i')))})^{\intercal} C_{i'}].
\end{aligned}$$

Because $i \neq i'$, by independence $\mathbb{E}[Z_i Z_{i'}] = \mathbb{E}[Z_i]\mathbb{E}[Z_{i'}] = \theta^2$. Because $i \notin \{i'\} \cup \mathcal{D}_{m(j(i))}$ and because $\mathbb{E}[C_{i'}] = 0$, we have

$$\mathbb{E}[Z_i (\hat{w}_n^{(m(j(i')))})^{\intercal} C_{i'}] = 0.$$

Similarly,

$$\mathbb{E}[Z_{i'} (\hat{w}_n^{(m(j(i)))})^{\intercal} C_i] = 0.$$

Finally notice that either $j(i) \neq m(j(i'))$ or $j(i') \neq m(j(i))$. Therefore,

$$\mathbb{E}[(\hat{w}_n^{(m(j(i)))})^{\intercal} C_i (\hat{w}_n^{(m(j(i')))})^{\intercal} C_{i'}] = 0.$$

We conclude that $\mathrm{Cov}(U_i, U_{i'}) = 0$.

We of course have $\hat{w}_n^{(j)} \to_p w^*$ for each $j = 0, 1, 2$. So, by multivariate CLT and Slutsky's theorem we have

$$\begin{pmatrix} \sqrt{|\mathcal{D}_1|} \left( \mathbb{E}_{D_1} Z - (\hat{w}_n^{(m(1))})^{\intercal} \mathbb{E}_{D_1} C - \theta \right) \\ \sqrt{|\mathcal{D}_2|} \left( \mathbb{E}_{D_2} Z - (\hat{w}_n^{(m(2))})^{\intercal} \mathbb{E}_{D_2} C - \theta \right) \\ \sqrt{|\mathcal{D}_3|} \left( \mathbb{E}_{D_3} Z - (\hat{w}_n^{(m(3))})^{\intercal} \mathbb{E}_{D_3} C - \theta \right) \end{pmatrix} \to_d \mathcal{N} \left( \begin{pmatrix} 0 \\ 0 \\ 0 \end{pmatrix}, \begin{pmatrix} V^* & 0 & 0 \\ 0 & V^* & 0 \\ 0 & 0 & V^* \end{pmatrix} \right).$$

Since $\frac{|\mathcal{D}_j|}{n} \to_p \frac{1}{3}$, we therefore have by Slutsky's and the continuous mapping theorem that

$$\sqrt{n}(\hat{\theta}_n^* - \theta) = \sum_{j=0,1,2} \sqrt{\frac{|\mathcal{D}_j|}{n}} \sqrt{|\mathcal{D}_j|} \left( \mathbb{E}_{D_j} Z - (\hat{w}_n^{(m(j))})^{\intercal} \mathbb{E}_{D_j} C - \theta \right) \to_d \mathcal{N}(0, V^*),$$

yielding the second statement. $\qquad \square$

## C   Additional results on the MSLR-WEB30K data

In Fig. 1 of this Supplementary Text we show results comparing the proposed PICVs estimator with PI and wPI on the MSLR-WEB30K data (Qin and Liu, 2013), using a lasso-based regression model for the target policy. As in the corresponding plot in the main paper (where we used a tree-based predictor for the target policy), here we vary $M$, the number of available actions (documents) per slot, $K$, the number of slots (size of ranked lists), and the choice of metric that defines slate-level reward. Missing values for wPI (for sample size 1k) are due to excessive variance caused by the presence of outliers. Results are qualitatively very similar to what we obtained with tree-based models. See main paper for details.

## D   Additional results on the synthetic data

In Fig. 2 of this Supplementary Text we reproduce the results of Fig. 2 of the main paper (simulations on a non-contextual slate bandit problem) but for a wider range of sample sizes. As in Section 7 of the main paper, we observe that the multi (PICVm) variant can dominate the single (PICVs) variant over

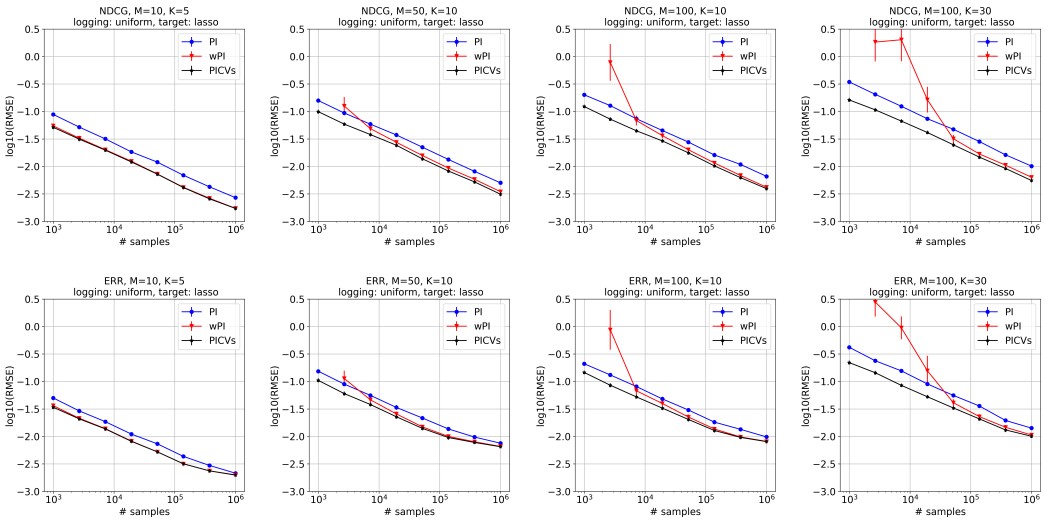

Figure 1: Benchmarking the proposed PICVs estimator against PI and wPI on the MSLR-WEB30K data, using a lasso-based regression model for the target policy.

a large range of sample sizes, albeit the improvement is not as pronounced as the improvement of PICVs over PI and wPI as shown in Fig. 1 of the main paper (and as we discuss in the first paragraph of Section 4.2 "Optimal Multiple Control Variates" of the main paper). For larger values of $K$ (we tried $K = 5$ and $K = 10$), PICVs and PICVm start demonstrating similar behavior (as in the experiment with the MSLR-WEB30K data described in Section 6 of the main paper), and we omit the corresponding plots.

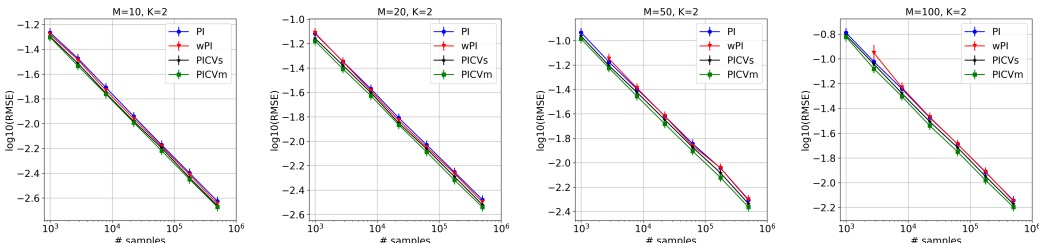

Figure 2: The analogous plots of Fig. 2 of the main paper (simulations on a non-contextual slate bandit problem) but here for a wider range of sample sizes. See text for details and discussion.

# References

Qin, T. and Liu, T. (2013). Introducing LETOR 4.0 datasets. *CoRR*, abs/1306.2597.