# OpenReview forum: "Control Variates for Slate Off-Policy Evaluation"
_NeurIPS.cc/2021/Conference — NeurIPS 2021 Poster_

### Official Review · Reviewer_AtDc · 2021-07-15

**Rating:** 6
**Confidence:** 4

**Summary:**

The paper proposes a new off-policy evaluation estimator for slate recommendation. The main idea focuses on a specific setup of Swaminathan et al, 2017, where the target / behavior policy are factorized. The new estimator generalizes the pseudo-inverse estimator by Swaminathan et al, 2017 with control variates, discusses asymptotic variance property of PI, self-normalized PI (wPI), and compares them empirically, showing that the new estimator based on minimizing the variance using control variate, achieves the best performance.

===== Updated post rebuttal ====

The authors have addressed my main technical concerns. I made quite some suggestions on the paper, such as adding some additional clarifications on the experiment results to the paper. If the authors are able to make such adjustment in the final manuscript, I am happy to stick with my original evaluation.

**Ethical Concerns:**

No clear ethical concerns.

**Limitations And Societal Impact:**

The author discusses some limitations and has included a societal impact section.

**Main Review:**

======== Originality ==========

Variance reduction using control variate is not a new idea for variance reduction in off-policy evaluation. However, I find that the connection between wPI and PI + control variate in the asymptotic case, to be rather interesting and original.

======== Quality ==========

The theoretical results of the paper seem to derive from fairly straightforward deduction, however, I still find them to be quite insightful and interesting. The empirical gains are a bit marginal.

========= Clarity ==========

The paper is clearly written in its theoretical part, but might require some further clarification in the empirical section.

========= Significance =========

This paper seems to make a new contribution to slate recommendation, by unifying a few ideas and suggesting new methods using control variates. I do have a few detailed tech questions below.

========== Detailed questions =========

1. The analysis is mostly based on asymptotics. And since pretty much all estimators are consistent, when sample sizes approach infinity, only the variance matters. However, overall, one might find such characterizations to be rather weak, given that arguably the main contribution of the paper is theoretical. Have the authors attempted finite-sample bias and variance analysis of variance estimators? In particular, for wPI, where there is a finite sample bias, I think it would be of interest to characterize the bias when the sample size is finite.

2. Once again, I find the connection between control variate + PI being equivalent to wPI to be quite interesting. The equivalence, however, is in the asymptotic case. It would be nice to have a finite sample characterization of the bias of wPI. Can the authors comment on this?

3. In Sec 5, a new estimator with unbiased mean and asymptotically optimal variance property is suggested. If I understand correctly, the intuition is to partition the data set into K folds and compute the control variate coefficient of one fold using other K-1 folds. This preserves the unbiasedness of the overall estimator. I think it might be better to present the general K fold case (instead of K=3 in the paper), and discuss the effect of K.

It would also be nice to have some more discussion on the intuition of this approach, before diving into the pseudo code.

4. What happens in Fig 1 when the sample size is 10 or 100? I wonder if the bias of wPI would be pretty hurtful and so is the variance of PI.

5. In Fig 2 middle and right plot, it seems that wPI performs worse than PI for pretty much all sample size. Is this contradictory to the claim that wPI has lower asymptotic variance?




**Time Spent Reviewing:**

3

---

> ### Author Response · Authors · 2021-08-09
> **Response to Reviewer AtDc**
>
> Thank you for taking the time to review our work and for your helpful feedback. We are glad you found the work interesting and original. Your questions are very much appreciated and will help us improve clarity. We answer these below in order:
>
> Q: Have the authors attempted finite-sample bias and variance analysis of variance estimators? In particular, for wPI, where there is a finite sample bias, I think it would be of interest to characterize the bias when the sample size is finite.
>
> A: Thanks for the question. We’d like to first highlight that wPI is not our focus and instead our focus is our new control-variate-based estimators; we only comment that asymptotically wPI behaves as one of these, but indeed there is some finite sample error there but it vanishes faster than $1/\sqrt n$. At the same time, we’d like to emphasize that lemma 11 establishes that our three-way cross-fitting estimator has zero bias even in finite samples. We could in fact adapt our cross-fitting procedure to obtain alternative estimators that are both unbiased and match the asymptotic variance of the single-control-variate version or even wPI. We need only change the weights computed in step 2. We’ll comment on this. To summarize: we can completely avoid bias when the sample size is finite. The variance characterization in all of our results is indeed asymptotic and we do not currently have corresponding finite-sample results. As this is an interesting question, we will comment on this in Section 8 and how such might be obtained from finite-sample results on estimating the optimal weights. We have experimentally observed that when the sample size is small, wPI is indeed behaving rather poorly (see Figures 1 and 2, and those in the Supplementary Text). This is due to wPI being defined as a ratio of estimates, and therefore the estimator is very sensitive to values of the denominator close to zero, but we do not have a precise result here for finite sample sizes.
>
> Q: Once again, I find the connection between control variate + PI being equivalent to wPI to be quite interesting. The equivalence, however, is in the asymptotic case. It would be nice to have a finite sample characterization of the bias of wPI. Can the authors comment on this?
>
> A: We first emphasize that although wPI is in the same family as our “PICVs”, the two estimators are not asymptotically equivalent as wPI corresponds to using a potentially suboptimal CV weight. (Lemma 7 establishes the gap.) The sense in which wPI is a control variate method is actually exact, not asymptotic, but the variance analysis is asymptotic. As the proof of lemma 4 shows, one can write wPI exactly as a single-control-variate estimator with the weight given by $\beta=E_n[GR]/E_n[G]$. Because the weights are data-driven, we study the asymptotic regime by considering their limit point. This is actually analogous to our estimators, which also use data-driven weights (but ones which converge to the optimal weights). So, to the extent that our estimators are “CV + PI,” so is wPI -- it is simply that our variance analysis of any “CV + PI” estimator using data-driven weights relies on asymptotic arguments (lemma 3).
>
> Q: In Sec 5, a new estimator with unbiased mean and asymptotically optimal variance property is suggested. If I understand correctly, the intuition is to partition the data set into K folds and compute the control variate coefficient of one fold using other K-1 folds. This preserves the unbiasedness of the overall estimator. I think it might be better to present the general K fold case (instead of K=3 in the paper), and discuss the effect of K. It would also be nice to have some more discussion on the intuition of this approach, before diving into the pseudo code.
>
> A: Thank you for these suggestions, we will revise the text accordingly and use generic K. Note that we do need K>=3 in order to ensure these properties: this is crucial on the last line of the first page of the supplement. Essentially, at least 3 folds are needed so that, when considering two data points i and i’, either i is independent of the data used to fit the weights of i’ or i’ is  independent of the data used to fit the weights of i. Two folds would not be enough for this. We will add an intuitive explanation of this before diving into the pseudocode as suggested.
>
> Q: What happens in Fig 1 when the sample size is 10 or 100? I wonder if the bias of wPI would be pretty hurtful and so is the variance of PI.
>
> A: When the sample size is very small relative to M, the wPI estimator performs very poorly (see discussion above). Often it would produce values that are several orders of magnitude off the median (we have omitted such excessive values from the plots to avoid clutter). PI is more robust than wPI for small sample sizes. We will clarify this in the text.
>
> Q: In Fig 2 middle and right plot, it seems that wPI performs worse than PI for pretty much all sample size. Is this contradictory to the claim that wPI has lower asymptotic variance?
>
> A: In these plots wPI performs worse than PI because the sample sizes are still rather small. In the Supplementary Text we show the analogous plots for larger sample sizes, where the behavior of the two estimators matches the asymptotic behavior we predict. Note that our estimators do not suffer this bad behavior.

---

### Official Review · Reviewer_apca · 2021-07-15

**Rating:** 7
**Confidence:** 3

**Summary:**

The paper addresses the problem of bandit slate off-policy evaluation, and introduces a family of estimators using control variates which effectively trade-off the bias and variance for these algorithms. The authors show their family of estimators generalizes two commonly used estimators and provide experimental evidence of the ability of their method to out perform these estimators on real and synthetic datasets.

**Limitations And Societal Impact:**

The authors properly adressed the societal impacts of their work.

**Main Review:**

This is overall a well written solid paper. The idea of generalizing equation (1) by adding control parameters in the summation term in G seems like an obvious next step in such line of research (hence not an earth shattering innovation), but the execution of this  idea, and the resulting theoretical results is performed well and elegantly. The optimal estimators derived are very straightforward to compute and therefore I see potential for the method to be adopted in future works. The experiments are not exhaustive, but certainly enough to demonstrate the merit of the method. Additionally, the results in section 7 do a good job demonstrating the additional benefit of slate dependent variates.

A few questions\comments to the authors:
- I think my main confusion stems from how lemma 4 interplays with the optimal CV found in section 4. If I understand correctly, since \beta^* should converge to a fixed point, then asymptotically, PICV should always converge to wPI (which in turn converges to PI). Is this true? It seems to happen in Figure 1 and not in Figure 2. Could Lemma 4 be used to estimate the number of samples at which PCVI and wPI become very close given V_w, and see if that number roughly corresponds to where the estimators meet in Fig 1 and show it for Fig 2 as well (assuming in Fig 2 it doesn't happen for an unreasonably large number of samples, but it is odd that the PCVI curves are parallel to the wPI curves and the wPI curves never catch up.
- It is unclear exactly which estimator is used in the experiments - invoking Lemma 6 in line 208 makes it seem like it's the vanilla estimator, while the parentheses in line 223 makes it seem like the cross-fitted estimator of section 5. Either way it would be interesting to see a comparison between the two estimators to get a sense of how bad the variance is when using the vanilla estimator.


**Time Spent Reviewing:**

~4 hours

---

> ### Author Response · Authors · 2021-08-10
> **Response to Reviewer apca**
>
> Thank you for reviewing our submission and for your encouraging comments. We are glad you found it solid and well-written. Thank you as well for the helpful feedback, which we will use to further improve the clarity of the paper.
>
> Let us answer your questions in order:
>
> Q: how lemma 4 interplays with the optimal CV found in section 4. If I understand correctly, since \beta^* should converge to a fixed point, then asymptotically, PICV should always converge to wPI (which in turn converges to PI). Is this true? It seems to happen in Figure 1 and not in Figure 2.
>
> A: $\beta^*$ is an (unknown) constant, and the data-driven $\widehat{\beta_n^*}$ (from lemma 6) does converge (in probability) to $\beta^*$ so that, by lemma 3, it obtains the same performance. Thus, as lemma 7 establishes, asymptotically PICV should (in theory) dominate both PI and wPI. Lemma 4 shows that wPI has asymptotic variance $V_\theta$ -- that is, it is equivalent to using the weight $\beta=\theta$ on the single control variate -- and lemma 2 shows that PI has asymptotic variance $V_0$. Both of these variances are larger than $V^\dagger=V_{\beta^*}$. So, it is not true in general that PICV should converge to wPI (or that wPI should converge to PI), at least in the sense of having similar asymptotic distributions, unless certain special conditions hold so that these variances are equal. The estimators do appear similar for large $n$ in Figure 1 due to the nature of the particular dataset and the reported setup (choice of metric, target policy, and value of $K$; see Supplementary Text for different settings). In the Supplementary Text we show the analogous plot of Figure 2 for even larger sample sizes, where we observe a uniform gap between PICV and (w)PI.
>
> Q: Could Lemma 4 be used to estimate the number of samples at which PCVI and wPI become very close given V_w, and see if that number roughly corresponds to where the estimators meet in Fig 1 and show it for Fig 2 as well (assuming in Fig 2 it doesn't happen for an unreasonably large number of samples, but it is odd that the PCVI curves are parallel to the wPI curves and the wPI curves never catch up.
>
> A: Thanks for the interesting question. First, we highlight that our theory results are asymptotic so they do not currently permit the finite-sample comparison suggested. However, our results do support the observation of parallel curves that never catch up. Let us explain. Figures 1 and 2 are on a log-log scale (log(MSE) vs log(# samples)). Our theory predicts that, for large enough $n$, the MSE of the different estimators will be $V/n$ for different values of $V$. Since $\log(V/n)$ = $\log(V)-\log(n)$, we expect all methods to eventually appear linear in the log-log scale with the very same slope (hence parallel), but at different heights given by $V$ (hence never catch up). The fact that PICVm is always at the bottom (whether separated in Figure 2 or on top of other estimators in Figure 1) is consistent with lemma 10, which shows that PICVm attains the smallest asymptotic variance among all other estimators. We will point this out clearly in Sections 6 and 7.
>
> Q: It is unclear exactly which estimator is used in the experiments - invoking Lemma 6 in line 208 makes it seem like it's the vanilla estimator, while the parentheses in line 223 makes it seem like the cross-fitted estimator of section 5. Either way it would be interesting to see a comparison between the two estimators to get a sense of how bad the variance is when using the vanilla estimator.
>
> A: In all experiments we have used the estimators from lemmas 6 and 9. We will revise the text in line 223 to clarify this. We agree that it would make for an interesting additional experiment to empirically compare to the cross-fitted estimator from lemma 11, which we can add in the supplement. Theoretically, at least, lemmas 9 and 11 show that the variance is actually the same for the two estimators, asymptotically, so one does not suffer additional variance due to not cross-fitting.

---

### Official Review · Reviewer_nq6H · 2021-07-17

**Rating:** 7
**Confidence:** 3

**Summary:**

The paper discusses a family of control variates for the problem of slate off-policy evaluation, under the assumption that the logging policy is a factored policy. For the defined class of estimators, the optimal weights are derived and improvements over unweighted estimators are quantified. Using a 3-way sample split the optimal weights are learned. Asymptotic statistics of the estimators are studied.

**Limitations And Societal Impact:**

See above

**Main Review:**

The paper is pretty clearly written and the contributions are novel. I do not have any major complaints.

**Time Spent Reviewing:**

1

---

> ### Author Response · Authors · 2021-08-09
> **Response to Reviewer nq6H**
>
> Thank you for taking the time to review our submission. We were glad to read that you found the paper clearly written and the contributions novel. Indeed, we worked hard to present our methods and results clearly in an instructive manner.

---

### Official Review · Reviewer_EZVP · 2021-07-19

**Rating:** 7
**Confidence:** 3

**Summary:**

This paper studies off policy evaluation in batch contextual bandits for multi dimensional action aka "slates". The paper studies the estimation of a certain target parameter which is equivalent to the expected reward under the key assumptions of (1) a target logging policy and (2) a decomposition of the conditional reward in terms of additive latent functions of the context and the action dimension. The authors consider control variates formed by linear combinations of the mean shifted importance weight ratios of each action dimension and show that a certain biased low variance previous estimator (wPI) can be recovered within this class. They also further derive minimum variance unbiased estimators from this class along with experimental evidence to support the theory.

**Limitations And Societal Impact:**

This paper is primarily a theory/ statistical techniques paper and only indirectly relevant for societal impacts.

**Main Review:**

The paper has novel results with clear exposition and technically sound proofs. Extensive experimental evidence is provided to demonstrate improvements of the new estimator over existing alternatives on several benchmark metrics. The authors also theoretically explore optimal estimators in the general class of weighted control variates considered and use a synthetic example to drill down into the somewhat surprising observation in the real data benchmarks that a single weight optimized version performs as well as the version with separate weights.

The assumptions needed could be considered technically strong (factored $\mu$ and additive rewards), but may be reasonable for certain settings. The authors claim that most of the estimation results hold generally without the additive decomposition of conditional reward; while this is technically true, it seems like the target parameter $E[GR]$ has significance primarily when this assumption holds. So this seems like a somewhat moot point.



**Time Spent Reviewing:**

3

---

> ### Author Response · Authors · 2021-08-09
> **Response to Reviewer EZVP**
>
> Thank you for reviewing our submission. We were heartened to read that you found the results novel, the exposition clear, the proofs technically sound, and the experiments extensive. We worked hard to present our methods and results clearly in an instructive manner. We agree on your latter point, and indeed our intent is to build on the pseudo-inverse estimator, which targets the $E[GR]$ estimand because it assumes the decomposition assumption. We only meant to highlight that our technical results are about estimation and not identification. We will further clarify this intent.

---

> > ### Comment · Reviewer_EZVP · 2021-08-29
> > **reply**
> >
> > Thanks for the reply and acknowledging the concern I had. I stick to my original review (which already had a high score)

---

### Decision · Program_Chairs · 2021-09-27

**Decision:**

Accept (Poster)

**Comment:**

The paper studies off-policy evaluation with combinatorial action spaces, referred to as "slate recommendation," and developed control-variate approaches that tradeoff bias and variance and provide a more complete story of the estimator landscape in this space. In addition, experimental results are provided, which help bridge the gap between theory and practice.

Overall, the paper is clearly written and well executed and the reviewers found few weaknesses. As such, we recommend acceptance. Congrats!